# Human Health Risk Assessment from the Tilapia Fish in Heavy Metal–Contaminated Landfill Reservoir

**DOI:** 10.3390/ijerph22060873

**Published:** 2025-05-31

**Authors:** Ni Yang, Pansa Monkheang, Lamyai Neeratanaphan, Somsak Intamat, Bundit Tengjaroensakul

**Affiliations:** 1Toxic Substances, Microorganisms and Feed Additives in Livestock and Aquatic Animals for Food Safety Research Program, Khon Kaen University, Khon Kaen 40002, Thailand; ni.y@kkumail.com; 2Faculty of Veterinary Medicine, Khon Kaen University, Khon Kaen 40002, Thailand; 3Yasothon Agricultural Research and Development Center, Department of Agriculture, Ministry of Agriculture, Yasothon 35000, Thailand; pansa.3544@gmail.com; 4Faculty of Science, Khon Kaen University, Khon Kaen 40002, Thailand; hlamya@kku.ac.th; 5Thatphanom Crown Prince Hospital, Nakornphanom 48110, Thailand; suwanwisit5@gmail.com

**Keywords:** aquatic, carcinogenic risk, DNA, hazardous chemicals, oxidative process

## Abstract

This study highlights the significant environmental and health risks associated with heavy metal contamination (As, Cd, Cr, and Pb) in *Oreochromis niloticus* (Nile tilapia) from two locations: the Khon Kaen municipal landfill (study site) and the Thapra commercial fish farm (reference site). It also evaluates potential human health risks and investigates genotoxicity and oxidative stress markers, including malondialdehyde, hydrogen peroxide (H_2_O_2_), catalase (CAT), and superoxide dismutase (SOD) in fish. Heavy metal concentrations were analyzed using inductively coupled plasma optical emission spectrometry. To determine genetic differentiation, inter-simple sequence repeats with dendrogram construction and genomic template stability (%GTS) were applied. The results showed that the average concentrations of As, Cd, Cr, and Pb in water samples were 0.0848, 0.536, 1.23, and 0.73 mg/L, respectively. These values exceeded safety limits, and the average Cd in sediment (1.162 mg/kg) was above regulatory thresholds. In fish muscle, the average metal concentrations (mg/kg) followed the order Cr (1.83) > Pb (0.69) > Cd (0.096) > As (0.0758), with Pb exceeding food quality standards. The bioaccumulation factor ranked as Cr > Pb > As > Cd. Health risk assessments, including health risk index and carcinogenic risk, suggested Pb contamination poses significant health risks through fish consumption. From dendrogram results, the %GTS of *O. niloticus* from the landfill and reference sites were 46.34 to 71.67% and 87.34 to 96.00%, respectively. This suggests that fish from the landfill site exhibited greater genetic diversity compared to those from the reference site. Specific oxidative stress markers revealed higher levels of H_2_O_2_ and significantly lower activities of CAT and SOD in landfill *O. niloticus* than in the reference site. These results emphasize the urgent need for environmental monitoring, stricter pollution controls, and improved waste management strategies to protect aquatic ecosystems and human health.

## 1. Introduction

One of the most difficult problems facing local Thai government authorities is solid waste management. Thailand produced 25.7 million tons of solid waste in 2022, of which 5.4 million tons, or roughly 21%, were disposed of in an unsanitary manner, primarily in open dump landfills, according to the Pollution Control Department (PCD) [1]. Waste mismanagement is prevalent due to factors like rapid population growth, lack of proper recycling and treatment facilities, insufficient landfill space, illegal dumping, lack of public participation, and decreased government funding. The primary disposal method remains landfilling, but inadequate waste management has resulted in toxic leachate seeping into the surrounding environment [2]. At the Khon Kaen municipal landfill adjacent to the waste-to-energy power plant, high concentrations of major heavy metals (HMs) such as As, Cd, Cr, and Pb have been detected in both leachate and local organisms [3,4,5,6,7]. The reference site in the Thapra commercial fish farm is situated in the green area of Khon Kaen city, without any industries or factories nearby. Swangneat et al. (2023) reported Cd, Cr, and Pb contaminations in the muscle, gills, and liver of cage-cultured *Oreochromis niloticus* from a landfill area, with only Cd level exceeding the safety standard by 13% [3]. Notably, this pollution extends beyond landfill sites and can have extended environmental impacts [5].

The accumulation of HMs in fish can have harmful health effects because these metals may be transferred to the human body upon consumption [8]. The severity of health impacts depends on the quantity of fish consumed [9]. Therefore, assessing human health risks is crucial when evaluating the potential effects of fish consumption in this area. This study aimed to analyze HM concentrations in water, sediment, and fish near the landfill. Additionally, it examined the bioaccumulation factor (BAF) of HMs in fish and the potential health risks associated with their consumption. The findings of this research can raise awareness among consumers and authorities about the possible hazards of fish consumption and contribute to future improvements in landfill environmental management.

*Oreochromis niloticus* is a freshwater fish that primarily inhabits rivers, reservoirs, and lakes. As an herbivorous species, it feeds on algae and aquatic plants. The fish is highly susceptible to heavy metal pollution and other environmental contaminants, making it a key bioindicator of environmental quality [10]. To assess pollution levels, monitoring HM concentrations in water and sediments is essential. Additionally, tracking fish health provides further evidence of HM contamination. HMs enter aquatic organisms through two primary pathways: direct ingestion of water and biota or non-dietary absorption via the gills and skin [11]. Research has shown that HM exposure in fish could lead to chromosomal damage and alterations in DNA structures [12,13]. Moreover, HMs induce oxidative stress by disrupting the balance between free radical production and elimination [14]. Beyond their direct impact on fish, HMs accumulate in aquatic organisms and transfer through the food chain to predators such as birds, animals, and humans, amplifying their harmful effects on the biosphere. To date, reports on the toxic effects of HMs on *O. niloticus* health and molecular genetics have been limited. Authors hypothesized that HMs contaminated in the landfill reservoirs could affect *O. niloticus* fish at the genetic level and oxidative phenomenon, and consumption of the fish could pose a carcinogenic risk to human health. Therefore, this study aims to measure the concentrations of As, Cd, Cr, and Pb in water, sediments, and *O. niloticus*, assess potential human health risks, and investigate genetic toxicity and oxidative stress biomarkers in *O. niloticus* fish following exposure to HMs in a landfill site.

## 2. Materials and Methods

### 2.1. Study Sites

The study site was located in the municipal landfill, Kham Bon village, Khon Kaen Province, Thailand. The reference site is an area where there is no contaminated leachate from industries, farms, or landfills. Geographic coordinates of the landfill and reference sites are at latitude: 16.59608238433, longitude: 102.806853323728 and latitude: 16.334442760000, longitude: 102.79644053868, respectively (Figure 1). The water, sediment, and *O. niloticus* samples were collected from landfill heavy metal–contaminated sites and compared to samples from a non-heavy metal–contaminated reference site in a commercial fish farm, Thapra subdistrict, Muang Khon Kaen, Khon Kaen Province.

### 2.2. Sample Methods

Water, sediment, and *O. niloticus* were randomly selected from each of the 5 spots at the landfill and the reference sites for heavy metal analysis, genetic analysis, and oxidative stress analysis (n = 5; five samples of each analysis) [4,5].

The sample size of *O. niloticus* in this study was limited due to a combination of practical and ecological constraints. First, the population of fish inhabiting the natural reservoir near the landfill site was relatively small, and ethical considerations required that authors avoid over-sampling to prevent further disruption to the ecosystem. Additionally, constraints such as limited access to the site, available resources, and the seasonal availability of fish influenced our sampling capacity. Despite the small sample size, care was taken to ensure the samples were representative and provide valuable insights that the study followed accepted scientific publications [4,5]. By outlining limitations, future research with a larger sample size and across multiple seasons or sites is recommended to validate and expand upon these findings.

The details of the sampling collection processes were as follows [4]: water samples were collected from the landfill and the reference sites in 1000 mL plastic bottles using the grab sampling method, 15–30 cm below surface water. Five water samples were taken from five randomly selected points and stored in polyethylene bottles. To preserve the samples, 1–2 drops of concentrated nitric acid were added to achieve a pH of approximately 2 and were stored at 4 °C before being analyzed to measure the heavy metal concentrations.

For sediment samples, they were collected using a grab sampler from the bottom of the landfill reservoir at a depth of approximately 40–50 cm below the water surface. Five samples were taken from five randomly selected points and stored in clear plastic bags. The samples were air-dried before further analysis to measure the heavy metal concentrations.

In this study, samples of 5 tilapia fish per site with an average weight of 62–85 g and aged approximately 3 months old were collected from each water site using the cast net fishing method. The fish were humanly euthanized with an overdose of tricane methane sulfonate (MS-222, Sigma-Aldrich, St. Louis, MO, USA). The tail muscle part was excised, dried in a hot air oven at a temperature of 60 °C for 5 days, and kept at room temperature [4,5] before the measurement of heavy metal concentrations. The gill tissue was preserved in pure ethanol, DNA was extracted and amplified, and its %GTS was detected. The blood serum of each fish was kept in a freezer at −80 °C before measuring the levels of oxidative stress biomarkers MDA, H_2_O_2_, SOD, and CAT.

### 2.3. Analysis of Water Quality

An analysis of water quality was conducted by comparing samples from a Khon Kaen landfill reservoir with the Thapra commercial fish pond as a reference site. The parameters assessed included temperature, total dissolved solids, dissolved oxygen, electrical conductivity, and pH levels (Table 1).

### 2.4. Measurement of Heavy Metal Concentration in Water

A total of 1.25 mL of 30% HNO_3_ was mixed into the 25 mL water sample and heated on a hot plate to a temperature of 95 ± 5 °C for 60 min. After cooling, deionized water was added to bring the volume up to 25 mL. The solution was filtered using filter paper, and the metals were analyzed using induction coupled plasma-mass spectrometry (ICP-OES, Perkin Elmer Optima 7000 DV, Shelton, CN, USA) [15]. HM concentrations in water samples were evaluated and compared according to Thailand’s surface water quality standards [16].

### 2.5. Measurement of Heavy Metal Concentration in Sediment

A total of 5 mL of nitric acid was mixed with 10 mL of hydrogen peroxide and 15 mL of sulfuric acid to an exact 1 g of sediment and heated on a hot plate at 95 ± 5 °C for 2 h. After cooling, the volume was filtered through No. 42 cellulose filter paper and increased to 50 mL using deionized water. The final samples were subjected to the same protocol for ICP-OES analysis, together with a standard reference material and a blank [17]. HM concentrations in sediment samples were evaluated and compared according to Thailand’s soil quality standards [18] and USEPA [19].

### 2.6. Measurement of Heavy Metal Concentration in O. niloticus

A total of 1 g each of nitric acid and hydrochloric acid were mixed with each *O. niloticus* fish tail muscle sample (n = 5) and put on a hot plate at 60 °C for 30 min. Then, 10 mL of hydrogen peroxide was added and placed on a hot plate at 95 ± 5 °C for 1 h. After cooling, deionized water was added to make 25 mL. The mixture was filtered through filter paper No. 1 and examined with ICP-OES [20]. HM concentrations in *O. niloticus* samples were evaluated and compared according to Thailand’s Ministry of Public Health’s standards [21].

### 2.7. Quality Control and Quality Assurance

Every 10 to 20 samples, across all tested samples, were evaluated in relation to quality control standards and method blanks (MBs). During the preparation and digestion stages, the concentrations of HMs found in the method blanks were deducted from those found in the test series samples. The laboratory fortified matrix (LFM) verified the accuracy of the analysis. The limits of detection (LOD) for As, Cd, Cr, and Pb were 0.001, 0.0001, 0.0002, and 0.001 mg/L, respectively. For the As, Cd, Cr, and Pb analyses, the ICP-OES wavelengths were 188.979, 226.502, 267.719, and 220.353 nm, respectively [22]. The accuracy of the heavy metal concentration results was evaluated with the APHA method [23]. The value of recovery of metals was determined based on acceptance criteria that fall within the range of 85% to 115% [19].

### 2.8. Bioaccumulation Factor (BAF)

BAF indicates how much metal an organism accumulates through uptake from its environment. The BAF can be calculated using the equation, where Cm represents the concentration of HMs in soil or water, and fish represents the concentration of HMs in fish based on wet weight [7].BAF = C tissue/Cm

Concentrations of HMs in fish muscle are expressed as Cm, while fish tissue concentrations (C tissue) are measured based on their wet weight.

### 2.9. Estimated Daily Intake (EDI)

Three factors were taken into account by the EDI when evaluating exposure to HMs through fish consumption: the average body weight of a Thai person, which was assumed to be 58 kg/person (Cluster and Program Management Office, National Science and Technology Development Agency, Pathum Thani, Thailand, 2024); the daily quantity of fish consumed (D); and the concentration of metal in the fish (C).

The EDI is calculated using the formula below [5].EDI = (C metal × W fish)/Bw

Normal fish eaters consume 0.0215 kg of W fish per day [24].

Consequences of acceptable limits HMs in life were reported by a joint FAO/WHO expert committee on food additives (September 2002), which were 0.01 mg/kg for As and 2mg/kg for Cd in freshwater non-predatory fish.

### 2.10. Health Risk Index (HRI)

The HRI was computed using the formula of the ratio of the oral reference dose (RfD) to the EDI for each metal for non-cancerous oral exposure [25].HRI = EDI/RfD
where 0.0003, 0.001, 0.003, and 0.004 mg/kg/day were the oral reference doses for As, Cd, Cr, and Pb, respectively [19].

### 2.11. Carcinogenic Risk (CR)

The following formula was used to determine the CR [26]:CR = EDI/CSFo
where As and Pb had oral carcinogenic slope factors (CSFo) of 1.5 mg/kg/day and 0.0085 mg/kg/day, respectively [19].

### 2.12. DNA Extraction and PCR Analysis

The DNA molecules extracted from the gills and livers of the *O. niloticus* were effectively amplified using 13 inter-simple sequence repeat (ISSR) primers as detailed in Table 2 in a PCR cycler (Flex Cycler^2^, Analytikjena, Jena, Germany) [23]. Each DNA band was assessed and documented using the following diallelic coding system: presence was denoted as 1, while absence was indicated as 0. The statistical analysis of the genotoxicity study involved transferring each assessed DNA band to the dendrogram configuration utilizing the NTSYSpc 2.1 software [27]. The results of all assessed bands were incorporated into the dendrogram to evaluate the genetic similarity of *O. niloticus* as in other fish species from the pollution sites (gold mine, electronic waste) under investigation [4,5,27]. All assessed bands’ results were examined in order to determine the %GTS and create a dendrogram [28].%GTS = (1 − a/n) × 100
where n is the total number of bands in the samples from the reference area; and a is the number of polymorphic bands found in the municipal landfill samples, which is equal to the sum of the appearance of a new band and the disappearance of a normal band.

### 2.13. Oxidative Stress Biomarkers

#### 2.13.1. Malondialdehyde (MDA)

The measurement of malondialdehyde (MDA) involved assessing thiobarbituric acid reactive substances. For this, a mixture of 10% TCA, 5 mM EDTA, 8% SDS, and 0.5 μg/mL BHT was combined with 150 μL of plasma. After adding 0.6% TBA, the mixture was boiled, cooled, and centrifuged, and the absorbance at 532 nm was measured. A standard curve was created using 1,1,3,3-tetraethoxypropane concentrations of 0.3–10 μM [29].

#### 2.13.2. Hydrogen Peroxide (H_2_O_2_)

Ammonium molybdate was used to measure the H_2_O_2_ concentration. An amount of 100 µL of 32.4 mM ammonium molybdate was combined with 100 µL of the supernatant sample. A UV-visible spectrophotometer was used to measure the absorbance at a wavelength of 405 nm following a 10 min incubation period [30].

#### 2.13.3. Superoxide Dismutase (SOD)

SOD was calculated by reducing nitro blue tetrazolium (NBT) using a xanthine oxidase system. When crude SOD was not present, a 100% reduction in NBT was found. A total of 25 µL of the enzyme extract sample, 100 µL of 50 mM potassium phosphate buffer (pH 7.8), 25 µL of 2.5 mM xanthine, and 5 µL of 20 mM NBT were all included in the reaction mixture. The reaction was then initiated by adding xanthine oxidase. At a wavelength of 550 nm, a UV-visible spectrophotometer measured the reduced NBT as a blue formazan product by superoxide anion. To provide the rate of NBT reduction at 0.025 A_550_/min (A_0_), the proper quantity of xanthine oxidase was added. A formula [A_0_-A_SOD_)/A_0_] × 100 was used to calculate the percentage of inhibition of NBT reduction in the rate of NBT reduction in the presence of crude SOD (A_SOD_) [29,30].

#### 2.13.4. Catalase (CAT)

Hadwana and Alib determined that CAT was measured by a lower H_2_O_2_ concentration. The amounts of 25 µL of enzyme extract, 150 µL of 50 mM potassium phosphate buffer (pH 7.0), and 25 µL of 100 mM H_2_O_2_ were all included in the reaction mixture. Using a UV-visible spectrophotometer set to 240 nm, the CAT activity was determined [31].

### 2.14. Statistical Analysis

The human health risk assessment (BAF, EDI, HRI, CR) was calculated according to the mentioned formula. %GTS and dendrogram construction were used to assess the genetic differentiation of *O. niloticus*. The concentrations of HMs in water, sediment, and *O. niloticus* samples and oxidative stress levels were analyzed and evaluated using an independent *t*-test in SPSS. The statistical analyses were performed at a confidence level of 95%.

## 3. Results

### 3.1. Water Quality

Indicators of water quality in the reference and landfill sites are presented in Table 3.

Temperature, total dissolved solids, dissolved oxygen, electrical conductivity, and pH from the landfill and reference site were within Thailand’s surface water quality standards [32].

### 3.2. Heavy Metal Concentrations in Water and Sediment

The concentrations of HMs present in the water and sediment from both the reference and landfill sites are respectively presented in Table 4 and Table 5. The concentrations of As, Cd, Cr, and Pb in water and the concentration of Cd in sediment from landfills exceeded the standard of the Pollution Control Department of Thailand [16,18]. Statistical analysis showed a significant difference between the Cd, Cr, Pb in water and As, Cd, Cr, and Pb in sediment from reference and landfill sites. 

### 3.3. Heavy Metal Concentrations in O. niloticus

The concentrations of HMs in *O. niloticus* from the reference and landfill sites are presented (Table 6). The concentration of Pb in *O. niloticus* exceeded the standard for contaminants in food according to the Notification of the Ministry of Public Health of Thailand [21]. Statistical analysis showed a significant difference between the As, Cd, Cr, and Pb concentrations in *O. niloticus* from reference and landfill sites.

### 3.4. Fish Sample’s BAFs of Heavy Metals

The fish BAF of HMs is shown in relative order as Cr > Pb > As > Cd (Table 7). BAF values greater than 1 were found in the fish cases of Cr when the BAF was calculated for water.

### 3.5. Possible Hazards to Human Health from Consuming Fish That Contains Heavy Metals

EDI, HRI, and CR were used to assess the possible health risks in the study population resulting from HM exposure to eating *O. niloticus*. From our findings in Table 8, the order of EDI values for HMs consumed by *O. niloticus* was Cr > Pb > Cd > As. The order of HRI values for HMs consumed by *O. niloticus* was Cr > As > Pb > Cd. The HRI values of all *O. niloticus* samples did not exceed 1, indicating that heavy metals (As, Cd, Cr, and Pb) might not cause serious health problems in this study area. The CR value of Pb was higher than 1 × 10^−4^, indicating that *O. niloticus* caught near landfills may accumulate Pb and may pose a cancer risk factor. The assessment of the HRI and CR indicated potential human health effects from As, Cd, Cr, and Pb via the consumption of *O. niloticus*.

### 3.6. Genotoxicity

The application of 13 ISSR primers results in the generation of a total of 621 bands, including 83 distinct characteristics with 27 similar band profiles and 56 unique band profiles (Table 9). The genetic difference and subsequent dendrogram analysis separated *O. niloticus* into 2 clusters: landfill area and reference area (Figure 2 and Figure 3). The %GTS ranges of *O. niloticus* range from 87.34 to 96.00% for reference samples 1.1 to 1.5 and from 46.34 to 71.67% for landfill study samples 2.1 to 2.5 (Table 10).

### 3.7. Oxidative Stress Biomarkers

Table 11 presents the oxidative stress biomarkers data for *O. niloticus* samples. The oxidative biomarker levels revealed that the concentrations of MDA and H_2_O_2_ were greater in the landfill fish than in the reference fish, while the SOD and CAT values were lower in the landfill fish compared to the reference fish. Statistically significant differences were observed in the MDA, H_2_O_2_, SOD, and CAT levels of *O. niloticus* liver samples from the reference and landfill sites.

## 4. Discussion

### 4.1. Water Quality Parameters

This study showed that the parameter values of the reference site and the landfill site were within the range of the Thai surface water standards [34]. Monitoring water quality, including temperature, TDS, EC, DO, and pH, can support that these sites had a potential impact on the suitability of the aquatic ecosystem [34,35]. The lower dissolved oxygen value in the landfill site in this study could impact various biological processes, including the stress response, metabolism, reproduction, genetics, and health conditions of fish [36,37,38].

### 4.2. Heavy Metal Concentrations in Water, Sediment and O. niloticus

HM levels in the water exceeded Thailand’s surface water quality standards. Cd concentrations in sediment samples surpassed Thailand’s soil quality standards. HM concentrations were higher in sediment than in water, likely due to human activities and ecological factors around the landfill, which contribute to HM accumulation in sediments. Additionally, *O. niloticus* from the reference site exhibited higher HM concentrations than those from the landfill site. Notably, Pb levels (0.69 mg/kg) (Table 6) in *O. niloticus* exceeded Thailand’s food quality standards (0.5 mg/kg) [32]. Consequently, these HMs are deposited in sediments and accumulate in *O. niloticus*.

There are two main ways that HMs can enter aquatic organisms: either by direct ingestion of water and biota or by non-dietary absorption through the skin and gills [11]. According to research, fish exposed to HM may experience changes in DNA structure and chromosomal damage [12,13]. Furthermore, by upsetting the equilibrium between the generation and removal of free radicals, HMs cause oxidative stress [14].

Numerous studies have documented HM accumulation in water, sediment, and fish. Intamat et al. [39] also reported that As levels in landfill-associated fish species, such as *Barbonymus gonionotus*, *Raiamas tornieri*, *Anabas testudineus*, and *Oreochromis niloticus*, exceeded Thailand’s food quality standards. Additionally, Noudeng et al. [40] found that while leachate metal concentrations remained within international standards, Cd, Cr, and Pb levels in fish from a Laos landfill exceeded permissible limits. Sriuttha et al. [41] reported that Cr concentrations in *A. testudineus* and *R. tornieri*, along with Cd levels in *B. gonionotus* and *O. niloticus*, exceeded international standards set by organizations such as the FAO and USA. The basic principles of EU legislation on contaminants in food are laid down in Council Regulation 315/93/EEC: Food. This regulation contains contaminants and their unacceptable amounts from the public health viewpoint and, in particular, at a toxicological level. Maximum levels must be set for certain contaminants in order to protect public health. Maximum levels for certain contaminants in food are set in Commission Regulation (EU) 2023/915. Maximum levels in certain foods are set for the following contaminants: metals (Pb, Cd, Hg, As). Conversely, Aytekin et al. [42] observed the lowest accumulation of Cd, Cr, and Pb in fish muscle. Additionally, Sriuttha et al. [41] found that Pb levels in *R. tornieri*, *A. testudineus*, *O. niloticus*, and *B. gonionotus* surpassed international guidelines. Aly et al. [43] noted that Fe, Zn, Cu, Cd, and Pb levels in *O. niloticus* remained within WHO standards; however, Pb concentrations in water samples exceeded the Egyptian chemical standards (1993) at Ismailia Canal, Egypt. Waichman et al. [44] investigated HM concentrations (Cd, Mn, Fe, Ni, Cu, Zn, As, Cr, Pb, and Hg) in 11 fish samples from the Brazilian Amazon River and found that Cr and Hg levels in fish meat exceeded the Brazilian safety limit for human consumption.

### 4.3. BAFs of Heavy Metals in the Fish Samples

The BAF is more ecologically relevant as it accounts for environmental exposure. The relative order of BAF values for HMs in fish absorbed from water was Cr > Pb > As > Cd. When calculating the BAF for water, fish containing Cr exhibited a BAF value greater than 1. This suggests that Cr has the potential for bioaccumulation from the water. As can damage the integument, nervous system, and digestive organs [45]. Thitiyan et al. [46] reported that *Barbonymus gonionotus* can accumulate low concentrations of As, Cr, Cd, and Pb from both soil and water. Similarly, Sriuttha et al. [41] observed BAF values exceeding 1 for Cd, Cr, and Pb in *Anabas testudineus*, *Rasbora tornieri*, and *Oreochromis niloticus* in a reservoir near a municipal landfill. Vaseem et al. [47] conducted an experiment to examine the effects of Cr, Pb, and Cu contamination in water on their accumulation in the gills, skin, muscle, and liver of *Labeo rohita*. These findings highlight the potential health risks posed by HM accumulation in fish, particularly for consumers who regularly ingest contaminated fish. Thus, a comprehensive health risk assessment is crucial to ensure food safety by estimating HM levels in consumed fish [48,49,50,51]. Kumar and Sharma [52] reported that Cd can damage genetic material, impair development and fertility, and negatively affect the nervous system. Cr exposure can cause acute symptoms such as vomiting, abdominal pain, diarrhea, and stomach bleeding. Chronic exposure may lead to skin irritation, ulcers, osteoporosis, or cancer. The toxicity of Cr depends on its oxidation state, with Cr⁶^+^ being more harmful than Cr^3+^ [53,54]. Additionally, prolonged exposure to Cr can result in lung cancer and death. Pb can also cause significant harm, affecting the kidneys, intestines, hemoglobin production, brain, and nerve cells [55].

### 4.4. Possible Harmful Effects of Heavy Metals on Health via Fish Consumption

Heavy metal (HM) pollution poses health risks through fish consumption, a key food source in Thailand. As, Cd, Cr, and Pb are particularly hazardous metals [3]. The relative EDI values followed the order: Cr > Pb > Cd > As. All EDI values were below the provisional maximum tolerable daily intake levels for As (0.42 μg/kg/day), Cd (1 μg/kg/day), Cr (100 μg/kg/day), and Pb (3.57 μg/kg/day). This indicates no significant health risk from HM exposure through the sampled fish. For *O. niloticus* consumption, all detected HM levels had HRI values below 1, based on the oral reference dose, suggesting it is generally safe for the local population. However, Pb concentrations in *O. niloticus* exceeded the recommended food safety levels, warranting caution. HM contamination in Thailand’s Nam Phong River, which is located near a municipal landfill. Local residents frequently consume aquatic plants and animals from the river, leading to HM accumulation in their bloodstream [56,57,58]. The United States Environmental Protection Agency (USEPA) classifies As and Pb as carcinogenic based on their CR values. According to USEPA (2015), CR values exceeding 1 × 10^−4^ are generally considered unacceptable [59]. In this study, the CR value of Pb was above the unacceptable threshold, while the CR value of As remained below it. Specifically, Pb contamination in *O. niloticus* exceeded the acceptable limit, whereas As levels did not pose a significant risk. Even low-level Pb exposure can lead to subtle neurological changes and has been associated with various cancers, including those of the skin, lungs, liver, prostate, and bladder. Additionally, Pb toxicity has been linked to diabetes, neurological disorders, cardiovascular diseases, and reproductive issues [60,61]. Therefore, in this study area, local residents consuming *O. niloticus* from the Nam Phong River are at risk of developing cancer due to Pb contamination. The CR values suggest that consuming *O. niloticus* near the landfill site may pose significant health risks to the local population.

### 4.5. Genetic Differentiation

A total of 621 DNA bands were produced by the ISSR patterns of 13 successful primers, of which 56 bands were unique. The %GTS of each individual *O. niloticus* in the reference area and municipal landfill is displayed in Table 9. *O. niloticus* from the municipal landfill had a %GTS range of 46.34 to 71.67%, whereas *O. niloticus* from the reference area had a range of 87.34 to 96.00%. In this study, the dendrogram results categorized the fish DNA samples into two distinct groups according to the sampling locations examined in Figure 2 and Figure 3. The findings indicate that metal accumulations in fish from landfills may impact their genotoxicity, as evidenced by alterations in the DNA bands. In environments contaminated with HMs, the metal exposure can result in genotoxic effects in fish, characterized by various forms of DNA damage. These effects may include single- and double-strand breaks, disruptions in DNA repair mechanisms, oxidation of nucleobases, and the formation of DNA-protein cross-links [62,63,64]. Studies on the interaction of HMs with cell membranes demonstrated genotoxicity through multiple mechanisms. For instance, Jha et al. [65] reported that As induced DNA alterations in *Channa punctatus*. Cd has been linked to oxidative stress, ploidy changes, gene base oxidation, DNA damage, mutagenesis, deletions, and point mutations [66,67]. Cr has been shown to cause nitrogenous base alterations, DNA strand breaks, and the formation of protein-Cr-DNA adducts [68,69,70]. Pb has been associated with carcinogenic events, DNA damage, oxidative stress, changes in gene transcription, and increased mitogenesis [71]. The loss of DNA structural or functional integrity in exposed organisms could have adverse effects at both individual and population levels, particularly in terms of growth and reproduction [71,72,73]. In this study, experimental *O. niloticus* demonstrates potential as a valuable genotoxic indicator in aquatic ecosystems.

### 4.6. Oxidative Stress

Oxidative stress is the result of an imbalance between antioxidants and ROS products and affects DNA molecules and cellular enzymes [26]. Accumulation of HMs and metalloid ions in organisms induces free radicals such as superoxide, hydrogen peroxide, and hydrogen ions, which can have potential effects on the health of organisms [25,74]. In this study, the liver of fish will be measured for MDA, H_2_O_2_, SOD, and CAT as oxidative stress biomarkers to monitor the extent of oxidative damage to the fish liver by HMs. The levels of MDA, H_2_O_2_, SOD, and CAT in fish collected from the landfill site were significantly different (*p* < 0.05) from those in fish collected from the reference area (Table 10).

MDA is a major byproduct of polyunsaturated fatty acid peroxidation in cellular membranes, and its levels rise with increased free radical activity. Measuring MDA concentration provides insight into the extent of oxidative damage caused by toxic substances [75,76]. This study found that fish from the landfill site had higher MDA concentrations compared to those from the reference site. Research has shown that toxic substances can accumulate in organisms, leading to elevated MDA levels [25,77,78]. Thitiyan et al. [46] observed a significant increase in liver MDA levels in *Barbonymus gonionotus* after exposure to As, Cd, Cr, and Pb.

H_2_O_2_ is a crucial precursor to the hydroxyl radical (OH), the most harmful reactive oxygen species (ROS) within cells [26]. This study revealed that fish from the landfill site had higher H_2_O_2_ concentrations than those from the reference site. The accumulation of HMs in fish contributed to increased H_2_O_2_ levels. Thitiyan et al. [46] reported that *B. gonionotus* from a municipal landfill site exhibited higher H_2_O_2_ concentrations than fish from a reference site, with an increasing trend linked to elevated As, Cd, Cr, and Pb levels. Similarly, Cd accumulation in *Channa striata* led to higher H_2_O_2_ concentrations at the study site compared to the reference site [26]. Additionally, Cd (VI) exposure resulted in increased H_2_O_2_ levels in *Channa punctatus* [30,79].

CAT and SOD levels were lower in the liver of *O. niloticus* from the landfill site compared to those from the reference site. These enzymes play a crucial role in the antioxidant defense system of organisms, helping to counteract the harmful effects of free radicals on vital biomolecules and tissues [80]. Specifically, SOD catalyzes the dismutation of superoxide anions into H_2_O_2_, while CAT subsequently breaks down H_2_O_2_ into oxygen and water [81].

Weber et al. [82] reported that *Hoplias intermedius* exposed to As, Pb, and Ni from tin mining waste exhibited reduced SOD and CAT levels compared to reference fish. As oxidative stress biomarkers influenced by As, Cd, Cr, and Pb, the levels of SOD, CAT, and MDA in *Barbonymus gonionotus* from a reservoir near a municipal landfill could serve as indicators for monitoring metal- and metalloid-induced oxidative stress. Similarly, Fatima et al. [83] found that *Channa striata* and *Heteropneustes fossilis* accumulated Pb, Cd, Cr, and Ni from the Kali River in Northern India, showing decreased SOD and CAT levels compared to reference fish.

Additionally, variations in oxidative stress biomarkers among fish species can be influenced by habitat differences, water quality parameters, feeding habits, species-specific responses, and the concentration and toxicity of contaminants.

A review of the literature data show that HMs, particularly Pb, can easily be absorbed and harms fish due to several mechanisms and factors, including high bioavailability in water. Pb exists in various soluble forms (e.g., Pb^2+^ ions) that can easily be absorbed by fish through their gills and digestive systems. It has an affinity for biological molecules. Pb binds readily to proteins, enzymes, and cellular components, interfering with normal biological processes and making it harder for fish to excrete [84]. Pb also has an affinity for biological molecules. Pb binds readily to proteins, enzymes, and cellular components, interfering with normal biological processes and making it harder for fish to excrete [85]. HM concentrations are also impacted by environmental conditions, including pH levels, water hardness, and temperature. Lower pH increases metal solubility, enhancing absorption. Soft water has fewer competing ions (e.g., Ca^2+^, Mg^2+^), making heavy metals more bioavailable. Higher temperatures increase fish metabolism and uptake rates [86]. Pb is also persistent in the environment. Unlike some metals that can be naturally cycled or diluted, Pb is highly persistent in ecosystems, leading to continuous exposure for fish populations [87].

## 5. Conclusions

HMs (As, Cd, Cr, and Pb) in the water of the landfill reservoir exceeded Thailand’s surface water standards. Sediment samples also showed Cd concentrations above Thailand’s soil quality standards, while Pb levels in fish samples surpassed the country’s food safety limits. A health risk assessment using the HRI indicated that consuming local *O. niloticus* did not pose a cancer risk. However, the CR value of Pb could be a cancer risk factor. Heavy metals in the contaminated environment could induce genetic alterations in *O. niloticus* near the landfill. Fish from this site exhibited oxidative stress effects. As a sustainable management strategy for preserving aquatic ecosystems around a landfill, the findings of this study are being used to create laws, regulations, and enforcement guidelines in close cooperation between government agencies and to protect local livelihoods.

## Figures and Tables

**Figure 1 ijerph-22-00873-f001:**
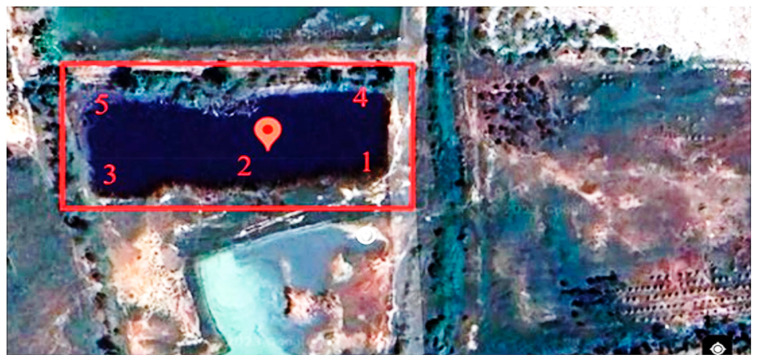
Red rectangle indicates the area where fish samples were collected from five different spots around the municipal landfill site, Kham Bon village, Muang District. Geographic coordinates of the landfill are latitude: 16.59608238433, longitude: 102.806853323728.

**Figure 2 ijerph-22-00873-f002:**
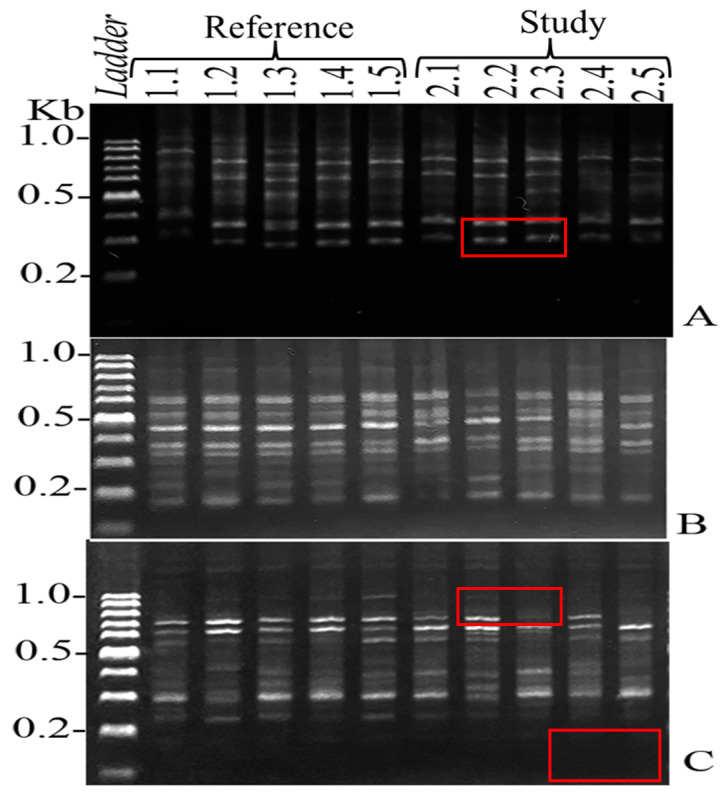
The following examples of ISSR fingerprints are presented for the reference (1.1, 1.2, 1.3, 1.4, and 1.5) and the landfill (2.1, 2.2, 2.3, 2.4, and 2.5) sites from the specific primers CACACACACACAAG (**A**), AGAGAGAGAGAGAGAAA (**B**) and AGAGAGAGAGAGAGAAC (**C**) showing monomorphic bands. Red rectangles present specific bands of target DNA.

**Figure 3 ijerph-22-00873-f003:**
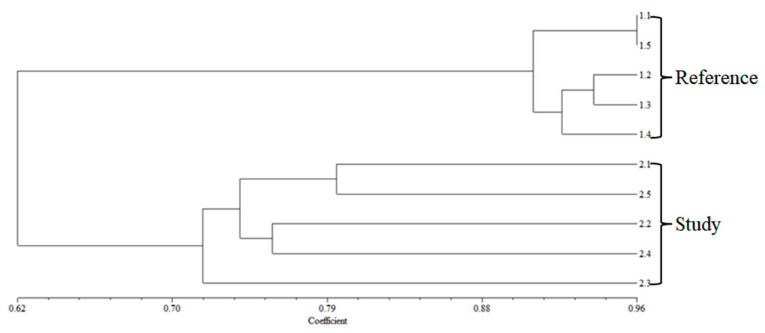
A dendrogram was generated using 13 primers through the NTSYSpc2.10 software, illustrating the genetic relationships among the *O. niloticus* samples, which include both the reference site (1.1–1.5) and the landfill site (2.1–2.5).

**Table 1 ijerph-22-00873-t001:** Analytical methods used for the analysis of water quality indicators.

Water Quality Indicators	Analytical Instruments
Temperature	Thermometer
Total dissolved solid	Total dissolved solids, Mettler Toledo, Model CH-8603
Dissolved oxygen	DO meter, Mettler Toledo, Model 966
Electro-conductivity	EC meter, Mettler Toledo, Model CH-8603
pH	pH meter, Eutech, Model EcoScan pH 5

**Table 2 ijerph-22-00873-t002:** The 13 primer sequences demonstrated successful amplification of ISSRs through polymerase chain reaction (PCR) in this investigation.

No.	Primer	Nucleotide Sequences
1	A4	AGAGAGAGAGAGAGAA
2	A11	AGAGAGAGAGAGAGAAA
3	A12	AGAGAGAGAGAGAGAAC
4	A14	AGAGAGAGAGAGAGAAT
5	P4	CACACACACACAAC
6	P5	CACACACACACAGT
7	P6	CACACACACACAAG
8	P7	CACACACACACAGG
9	P10	GAGAGAGAGAGACC
10	P12	CACCACCACGC
11	P13	GAGGAGGAGGC
12	P14	CTCCTCCTCGC
13	P15	GTGGTGGTGGC

**Table 3 ijerph-22-00873-t003:** The water quality indicators of the reference and the landfill sites.

	Indicators				
Samples	Temperature (°C)	TDS (mg/L)	DO (mg/L)	EC (µs/cm)	pH
Reference	31.36 ± 0.72	0.47 ± 0.03	6.57 ± 0.24	317.62 ± 6.08	7.65 ± 0.46
Landfill	29.06 ± 0.65	0.51 ± 0.06	4.51 ± 0.38	456.72 ± 4.88	7.16 ± 0.61
Standard	N/A	N/A	≥4.00	N/A	5–9

Remarks: TDS = total dissolved solids; DO = dissolved oxygen; EC = electro-conductivity; N/A= not available; (numbers of measured individuals) = 5; Thailand’s surface water quality standards [32].

**Table 4 ijerph-22-00873-t004:** The concentrations of heavy metals in the water samples from the reference site and landfill site (mean ± standard deviation; n = 5).

Concentration (mg/L)				
Study Site	As	Cd	Cr	Pb
Reference	0.0028 ± 0.0008	0.03 ± 0.006	0.015 ± 0.011	0.031 ± 0.007
Landfill	0.0848 ± 0.142	0.536 ± 0.139	1.23 ± 0.445	0.73 ± 0.19
*p*-value	0.268	0.001 *	0.004 *	0.001 *
Standard	0.01	0.05	0.05	0.05

Remarks: The Pollution Control Department (PCD) of the Ministry of Natural Resources and Environment of Thailand established standards for water quality in surface sources [16]. * Indicate a statistically significant difference from the pooled standard error (*p*-value).

**Table 5 ijerph-22-00873-t005:** The concentrations of heavy metals in sediment samples from the reference site and the landfill site (mean ± standard deviation, n = 5).

Concentration (mg/kg)				
Study Site	As	Cd	Cr	Pb
Reference	0.556 ± 0.236	0.0523 ± 0.126	12.64 ± 2.35	5.32 ± 0.97
Landfill	1.274 ± 0.436	1.162 ± 0.428	16.68 ± 2.08	14.19 ± 3.15
*p*-value	0.012 *	0.013 *	0.019 *	0.002 *
Standard	3.9	1	100	100

Remarks: The standard for soil quality, as established by the Pollution Control Department (PCD) of the Ministry of Natural Resources and Environment of Thailand [18]. * Indicate a statistically significant difference from the pooled standard error (*p*-value).

**Table 6 ijerph-22-00873-t006:** The concentrations of heavy metals in *O. niloticus* muscles sample from the reference site and the landfill site (mean ± standard deviation; n = 5).

Concentration (mg/kg)				
Study Site	As	Cd	Cr	Pb
Reference	0.0352 ± 0.0069	0.0272 ± 0.0024	1.17 ± 0.36	0.094 ± 0.08
Landfill	0.0758 ± 0.0173	0.096 ± 0.033	1.83 ± 0.09	0.69 ± 0.63
*p*-value	0.001 *	0.009 *	0.014 *	0.103 *
Standard	2	0.5	2.0	0.5

Remarks: The quality standards for food, as established by the Ministry of Public Health of Thailand [21]. * Indicate a statistically significant difference from the pooled standard error (*p*-value).

**Table 7 ijerph-22-00873-t007:** HM BAFs in fish samples.

	Heavy Metals	*O. niloticus*
BAF values based on water	As	0.893 ± 1.83
	Cd	0.179 ± 0.112
	Cr	1.487 ± 0.663
	Pb	0.945 ± 1.557

**Table 8 ijerph-22-00873-t008:** Heavy metals’ EDI, HRI, and CR through ingestion of *O. niloticus*.

Heavy Metals	EDI (μg/kg/day)	HRI	CR
*O. niloticus*			
As	0.028098	0.09366	0.000018732
Cd	0.035586	0.03558	-
Cr	0.678362	0.22612	-
Pb	0.255776	0.06394	0.03009
Unacceptable risk levels	-	>1 [20]	>1 × 10^−4^ [33]

**Table 9 ijerph-22-00873-t009:** The 13 successful primer sequences for ISSR-PCR fingerprinting with DNA band.

No.	Primer	Nucleotide Sequences	Total Bands	Monomorphic Band	Polymorphic Band
1	A4	AGAGAGAGAGAGAGAA	47	2	6
2	A11	AGAGAGAGAGAGAGAAA	73	1	8
3	A12	AGAGAGAGAGAGAGAAC	49	2	5
4	A14	AGAGAGAGAGAGAGAAT	68	3	5
5	P4	CACACACACACAAC	32	3	1
6	P5	CACACACACACAGT	36	1	5
7	P6	CACACACACACAAG	38	2	5
8	P7	CACACACACACAGG	18	1	1
9	P10	GAGAGAGAGAGACC	47	4	1
10	P12	CACCACCACGC	52	2	4
11	P13	GAGGAGGAGGC	43	1	5
12	P14	CTCCTCCTCGC	50	2	4
13	P15	GTGGTGGTGGC	68	3	6
		Total	621	27	56

**Table 10 ijerph-22-00873-t010:** The percentage of genomic template stability (% GTS) for the *O. niloticus* collected from the reference (1.1–1.5) and the landfill (2.1–2.5) sites.

					%GTS					
Fish No.	1.1	1.2	1.3	1.4	1.5	2.1	2.2	2.3	2.4	2.5
1.1										
1.2	87.65									
1.3	91.03	93.83								
1.4	87.34	92.59	91.25							
1.5	96.00	91.36	94.87	88.75						
2.1	61.84	61.73	60.00	62.82	63.64					
2.2	47.37	46.34	48.10	46.84	49.35	62.50				
2.3	63.16	60.98	61.25	62.03	60.76	63.49	61.40			
2.4	59.46	57.50	57.69	58.44	59.21	64.41	62.26	63.33		
2.5	67.57	65.00	65.38	64.10	69.33	71.67	61.40	60.00	63.33	00.00

**Table 11 ijerph-22-00873-t011:** Oxidative stress biomarkers values of *O. niloticus* fish from the reference site and the landfill site (mean ± standard deviation, n = 5).

Parameter	Reference	Landfill	*p*-Value
MDA (mmol/MDA/g liver)	31.11 ± 0.75	56.8 ± 8.93	0.003 *
H_2_O_2_ (mmol/g liver)	13.54 ± 0.906	20.65 ± 3.12	0.001 *
SOD (%)	33.45 ± 2.95	20.41 ± 6.51	0.004 *
CAT (Units/mg protein)	23.51 ± 2.96	14.85 ± 3.48	0.003 *

Remark: * A statistically significant difference (*p* < 0.05) is indicated by the pooled standard error (*p*-value).

## Data Availability

Data presented in this study are available on request from the corresponding author

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
