# Peer review of "Human Health Risk Assessment from the Tilapia Fish in Heavy Metal–Contaminated Landfill Reservoir"

_ijerph, 2025, doi:10.3390/ijerph22060873_

Round 1

Reviewer 1 Report

Comments and Suggestions for Authors

Journal: International Journal of Environmental Research and Public Health

Title: Human health risk assessment from the tilapia fish in heavy metal contaminated landfill reservoir

Comment: The manuscript investigates the health risks associated with heavy metal contamination in tilapia fish from landfill sites, focusing on specific metals (i.e., Cr, Pb, As, Cd) concentrations, bioaccumulation factors, and genetic analysis. The study presents valuable data, including heavy metal concentrations exceeding safety limits, genetic bands indicating differentiation among fish samples, and oxidative stress markers such as malondialdehyde (MDA) and catalase levels. However, several aspects require clarification, additional data presentation, and methodological rigor to enhance the manuscript's quality.

Abstract

  1. The abstract should quantify findings related to heavy metal concentrations. For example, provide specific values for cadmium (Cd), lead (Pb), chromium (Cr), and arsenic (As) in fish muscle, sediment, and water samples.
  2. Specify which biomarkers were used to assess oxidative stress and genotoxicity, such as the levels of malondialdehyde (MDA) and the activity of catalase and superoxide dismutase (SOD).

Introduction

  1. Clarify the consequences of heavy metal accumulation in fish, supported by recent statistics (e.g., mention acceptable limits for metals in aquatic life).
  2. Expand on the implications of inadequate waste management. Include recent statistics or studies to emphasize the severity of the problem.

Materials and Methods

  1. Line 97-100: Provide precise details on the number of fish sampled and the selection criteria for sampling sites. This detail is crucial for reproducibility.
  2. Incorporate the detection limits for the ICP-OES method used to analyze heavy metal concentrations, as this data is essential to evaluate the method's sensitivity and reliability.

Results

  1. Enhance the map to clearly indicate sampling locations with geographic markers (longitude and latitude).
  2. Figure 2: The current version of the figure is blurry. Please update it with a high-resolution version. Meanwhile, could you specify which band represents the target DNA? Please mark it clearly.
  3. HRI and CR values are critical; please specify the values for all detected heavy metals, particularly Pb, indicating how these values relate to safety thresholds.

Discussion

  1. Expand on the implications of elevated lead levels, particularly how the measured concentrations (e.g., Pb in muscle tissue exceeding the food safety standard of [insert specific value]) compare to existing literature.
  2. Provide quantitative analysis from the genetic dataset, such as the total number of bands detected and the percentage of unique versus common bands to illustrate genetic differentiation.
  3. Ensure consistency throughout the full text by using abbreviations appropriately. Do not mix full names and abbreviations.

Conclusion

  1. Summarize critical findings quantitatively and qualitatively, such as the specific health risks posed by Pb concentrations to local populations due to fish consumption. Explicitly mention the need for interventions in waste management practices.

References

  1. Include updated references from the last five years to underline the urgency of the situation regarding landfill waste management and its environmental impact.

Author Response

Abstract
The abstract should quantify findings related to heavy metal concentrations. For example, provide specific values for cadmium (Cd), lead (Pb), chromium (Cr), and arsenic (As) in fish muscle, sediment, and water samples. 
The results showed that the average concentrations of As, Cd, Cr and Pb in water samples were 0.0848, 0.536, 1.23 and 0.73 mg/L, respectively. These values exceeded safety limits, and average Cd in sediment (1.162 mg/kg) was above regulatory thresholds. In fish muscle, the average metal concentrations (mg/kg) followed the order Cr (1.83) > Pb (0.69) > Cd (0.096) > As (0.0758), with Pb exceeding food quality standards.
Page 1. Line 20-25.

2.Specify which biomarkers were used to assess oxidative stress and genotoxicity, such as the levels of malondialdehyde (MDA) and the activity of catalase and superoxide dismutase (SOD).
Biomarkers: Page 1.  Line 31-32.

Specific oxidative stress markers revealed higher level of H2O2 and significantly lower activities of CAT and SOD in landfill O. niloticus than in the reference site.
Genotoxicity:
From dendrogram results, the %GTS of O. niloticus from the landfill and reference sites were 46.34 to 71.67% and 87.34 to 96.00%, respectively. This suggests that fish from the landfill site exhibited greater genetic diversity compared to those from the reference site.
Genotoxicity:  Page 1.   Line 28-30.

Introduction
1.    Clarify the consequences of heavy metal accumulation in fish, supported by recent statistics (e.g., mention acceptable limits for metals in aquatic life).
Consequences of acceptable limits HMs in life were reported by joint FAO/WHO expert committee on food additives (September 2002) were 0.01 mg/kg for As, 2mg/kg for Cd, 0.3 mg/kg for Hg in freshwater non-predatory fish, and 0.6 mg/kg for Hg in freshwater predatory fish.
Page 2. Line 55-58.
2.Expand on the implications of inadequate waste management. Include recent statistics or studies to emphasize the severity of the problem.
One of the most difficult problems facing local Thai government authority is solid waste management. Thailand produced 25.7 million tons of solid waste in 2022, of which 5.4 million tones, or roughly 21%, were disposed of in an unsanitary manner, primarily in open dump landfill, according to the Pollution Control Department (PCD) [1]. Waste mismanagement is prevalent due to factors like rapid population growth, lack of proper recycling and treatment facilities, insufficient landfill space, illegal dumping, lack of public participation and decreased government funding.
Page 1. Line 38-44

Materials and Methods
1.    Line 97-100: Provide precise details on the number of fish sampled and the selection criteria for sampling sites. This detail is crucial for reproducibility. 
Edited in yellow highlight 
Number of fish: Page 3. Line:104-106. Added references[4-5]
Sampling sites: page2-3.  Figure 1. Line 92-102.

2.Incorporate the detection limits for the ICP-OES method used to analyze heavy metal concentrations, as this data is essential to evaluate the method's sensitivity and reliability. 
The limits of detection (LOD) for As, Cd, Cr and Pb were 0.001, 0.0001, 0.0002 and 0.001 mg/L, respectivelyFor the As, Cd, Cr, and Pb analyses, the ICP-OES wavelengths were 188.979, 226.502, 267.719, and 220.353 nm, respectively [23].    Page 4. Line 160-162.  

Results
1.    Enhance the map to clearly indicate sampling locations with geographic markers (longitude and latitude). Enhanced Figure 1 quality. Geographic coordinates of the landfill and reference sites are at latitude: 16.59608238433, longitude: 102.806853323728 and latitude: 16.334442760000, longitude: 102.79644053868, respectively.
      Sample location: Figure 1.Page 2-3. Line 92-102.
2.    Figure 2: The current version of the figure is blurry. Please update it with a high-resolution version. Meanwhile, could you specify which band represents the target DNA? Please mark it clearly. Figure 2.
Edited in Figure 2. Page 10.Line 315-318.

3.HRI and CR values are critical; please specify the values for all detected heavy metals, particularly Pb, indicating how these values relate to safety thresholds. Edited Figure 2.
The order of HRI values for HMs consumed by O. niloticus was Cr > As > Pb > Cd. The HRI values of all O. niloticus samples did not exceed 1 indicating that heavy metals (As, Cd, Cr, Pb) might not cause serious health problems in this study area. The CR value of Pb was higher than 1×10−4 indicating that O. niloticus caught near landfills may accumulate Pb and may pose a cancer risk factor. The assessment of the HRI and CR indicated potential human health effects from As, Cd, Cr, and Pb via the consumption of O. niloticus.  Page 8.   Line 297-303.

Discussion
Expand on the implications of elevated lead levels, particularly how the measured concentrations (e.g., Pb in muscle tissue exceeding the food safety standard of [insert specific value]) compare to existing literature.
Pb levels (0.69 mg/kg) (table 6) in O. niloticus exceeded Thailand's food quality standards (0.5 mg/kg).
Page 11-12. Line 353-354. Table 6. page 8. Line 285-286.
2.    Provide quantitative analysis from the genetic dataset, such as the total number of bands detected and the percentage of unique versus common bands to illustrate genetic differentiation.
The total number of bands detected in Table 9 
     Table 9, Page 9, Line 312-313.      
The percentage of unique versus common bands: Table 10

         Table 10, Page 11, Line 325-326.

3.Ensure consistency throughout the full text by using abbreviations appropriately. Do not mix full names and abbreviations. 
Edited in yellow highlight    Edited throughout the full text.

Conclusion
1.    Summarize critical findings quantitatively and qualitatively, such as the specific health risks posed by Pb concentrations to local populations due to fish consumption. CR value of Pb was higher than 1×10−4 indicating Pb could pose a cancer risk factor.    Page 15.    Line 520-521. 
2. Explicitly mention the need for interventions in waste management practices.
As a sustainable management strategy for preserving aquatic ecosystems around a landfill, the findings of this study are being used to create laws, regulations, and guidelines of enforcement with close cooperation between government agencies and to protect local livelihoods. Page 15. Line 523-526.

References
Include updated references from the last five years to underline the urgency of the situation regarding landfill waste management and its environmental impact.  Updated situation. Page 1, Line 38-44.
         Added reference [1]. Page 16, Line 547-548.

Reviewer 2 Report

Comments and Suggestions for Authors

This study addressed the difference in heavy metal load in fish, water and sediment of tilapia rearing lands in Thailand. The authors found interesting results and raised concerns about the environmental pollution and health health risks. The following issues should be resolved:

Abstract: It is generally well-written, but the following minor corrections are needed.

L16: here, state the name of the metal tested.

L23: indicate which genetic tests were conducted.

Introduction: The authors must address the current knowledge about the contamination of water, sediment and tilapia in Thailand and preferably the study site.

Methods:

Indicate which tilapia species were monitored in this study.

Sampling procedures must be in detail! How were the samples collected, preserved ....?

L85-89: the sentence should be in past form, not imperative

Indicate which part of the fish muscle were analyzed

L147-149: the methods of these tests must be briefly explained

Results: needs revisions

In the tables, change the term "standard" to "Thailand's surface water quality standards" or a defined abbreviation like TSWQS

In the table footnotes, not need to add "P<0.05", as the exact p-values were added to the tables, already

Discussion was well-written

Author Response

Question:
L16: here, states the name of the metal tested. 
Named As, Cd, Cr and Pb. Page 1, Line 13.

L23: indicate which genetic tests were conducted.
To determine genetic differentiation, inter simple sequence repeats with dendrogram construction and genomic template stability (%GTS) were applied. Page 1, Line 19-20.

Introduction:    The authors must address the current knowledge about the contamination of water, sediment and tilapia in Thailand and preferably the study site.    
Edited.    Page 2. Line 50-54. Added reference [3]. Page 16. Line552-554.

Materials And Methods:
Indicate which tilapia species were monitored in this study.
O. niloticus. For example, Page 3. Line 95-96. Line 100. Line 104.
Sampling procedures must be in detail! How were the samples collected, preserved ....?
Edited. Page 3. Line110-126.

L85-89:    the sentence should be in past form, not imperative    Edited in green highlight 
Page 4, Line133-138.

Indicate which part of the fish muscle were analyzed    
The tail muscle part. Page 3. Line 122.

L147-149: the methods of these tests must be briefly explained:
Added each method, includes: Malondialdehyde (MDA), Hydrogen peroxide (H2O2), Superoxide dismutase (SOD), Catalase (CAT). Page 6. Line 213-242.

Results: 
In the tables, change the term "standard" to "Thailand's surface water quality standards" or a defined abbreviation like TSWQS    Edited in green highlight Thailand's surface water quality standards.    
Edited. Page 7. Line 254-255. Line 258-259.

In the table footnotes, not need to add "P<0.05", as the exact p-values were added to the tables, already.    
Edited. Page7-8. Line 271-2

Reviewer 3 Report

Comments and Suggestions for Authors

this is a very adequate reseach task 

this is a very important research study

it is well written and well perforemed 

the study has one weak point: the description of the fish is missing ; how big, what age and and what condition factor 

These parameters are needed to be able to compare the metal loads in the fish as growth and age do determine the uptake. 
please add those data 

the rest of the methods and results is well performed 

comparisons to available limits in different countries are made. You could perhaps add the EU regulation data to.

About the legislation

The basic principles of EU legislation on contaminants in food are laid down in Council Regulation 315/93/EEC:

  • Food containing a contaminant to an amount unacceptable from the public health viewpoint and in particular at a toxicological level, shall not be placed on the market
  • Contaminant levels shall be kept as low as can reasonably be achieved following recommended good working practices
  • Maximum levels must be set for certain contaminants in order to protect public health

Maximum levels for certain contaminants in food are set in Commission Regulation (EU) 2023/915. Maximum levels in certain foods are set for the following contaminants:

  • metals (lead, cadmium, mercury, arsenic, inorganic tin)

Author Response

1.The study has one weak point: the description of the fish is missing; how big, what age and and what condition factor 
For fish O. niloticus sample (5 fish average 62- 85 g and age around 3 months), they were collected using the cast net fishing method, with a total of five fish per water site.    Page 3. Line 120-121.

2.These parameters are needed to be able to compare the metal loads in the fish as growth and age do determine the uptake. 
please add those data  Edited in pink highlight Page 2. Line 56-59.

3.Comparisons to available limits in different countries are made. You could perhaps add the EU regulation data to. 
Edited in pink highlight Page 12. Line 371-377.

Reviewer 4 Report

Comments and Suggestions for Authors

In this paper authors aimed to determine the concentrations of As, Cd, Cr, and Pb in water, sediment, and Nile tilapia (Oreochromis niloticus) from contaminated landfill reservoir as well as to compare this finding with Thailand legislations and assess the health risks posed by these elements in both fish and humans. Bioaccumulation factor was also calculated and biomarkers (genotoxicity and oxidative stress parameters) were also assessed. They found that all elements in water samples exceeded safety limits as well as Cd in sediment. According to the health risk index and carcinogenic risk analysis, it was suggested that levels of Pb could pose significant health risks due to fish consumption. Analyses of both biomarkers indicated greater changes in fish from contaminated site compared to the control. In conclusion, the authors pointed out the need for regular monitoring, pollution control, and improvement of waste management control.

The authors need to address the following comments before the paper get accept.

General comments

  • There is a lack of novelty in this paper.
  • This study is of a local character.
  • In Abstract, authors should indicate sampling site, analyzed heavy metals, and genotoxicity and oxidative stress parameters.
  • Keywords are to general and should be rewritten.
  • Write a general hypothesis in Introduction. In present form, it looks more like an essay than a scientific paper.
  • In Materials and methods, sampling methods for water, sediment, and fish are missing. The measurements of heavy metals in water, sediment, and fish are written as if they were copied from some manual. Please, rewrite this part and add the missing information.
  • Information on national standards for water, sediment, and fish meat should be defined in Materials and Methods, not in the Results. Also, add some international standards and legislations (e.g. FAO, USEPA, etc.).
  • Regarding the genotoxicity. I am not sure how this analysis indicates the pollution burden. I suggest using comet and/or micronucleus tests.
  • There is a lot of repetition of the results in the Discussion section. Please, avoid it.
  • Particular fish species were mentioned in the Discussion section. Do they have a similar ecology to Nile tilapia?
  • Rewrite the Conclusion section. It is just a repetition of the main findings.
  • In the main text, the authors should choose how to refer a species, Nile tilapia or niloticus. Mixing names loses consistency.

Specific comments

  • Line 106: Indicate which standard reference materials you used.
  • Line 120: Indicate Bw. Is this average body weight of adults in Thailand?
  • Line 132: Delete "CR stands for cancer risk".
  • Lines 151-152: Delete this sentence.
  • Line 159: Delete "(Table 3)" and write "in Table 3".
  •  Lines 206-207: Unclear. Rewrite this sentence.
  • Line 298: Delete this sentence.

Author Response

General comments 
Abstract
1.Authors should indicate sampling site, analyzed heavy metals, and genotoxicity and oxidative stress parameters.
Edited in blue highlight 
Sample site: Page 1. Line 12-15. 
Heavy metal analyzed: Page 1. Line 13, 17-18.
Genotoxicity: Page 1. Line 19-20.
Oxidative stress: Page 1. Line 16-17

2. Keywords general and should be rewritten. 
Edited in blue highlight aquatic; carcinogenic risk; DNA; hazardous chemicals; oxidative process Keywords  Page 1. Line 36.

3. Introduction Write a general hypothesis in Introduction. 
Edited in blue highlight Page 2. Line 82-85.

4. Materials and methods 
The Sampling methods for water, sediment, and fish are missing. 
Edited in blue highlight Page 3. Line 103-126. 
The measurements of heavy metals in water, sediment, and fish are written. Please, rewrite this part and add the missing information. 
Edited in blue highlight Page 4. Line 132-154.

Information on national standards for water, sediment, and fish meat should be defined in Materials and Methods, not in the Results. Edited in blue highlight Water standard: Page 4. Line 137-138.
Soil standard: Page 4. Line 144-146.
Fish standard: Page 4. Line 153-154.
Also, add some international standards and legislations (e.g. FAO, USEPA, etc.). Edited in blue highlight Page 4. Line 162-165.
Regarding the genotoxicity. I am not sure how this analysis indicates the pollution burden. I suggest using comet and/or micronucleus tests. 
Edited in blue highlight Page 5. Line 199-203. 
Repetition of the results in the Discussion section. 
Edited in blue highlightParticular fish species were mentioned in the Discussion section. Do they have a similar ecology to Nile tilapia? 
They don’t have a similar ecology to Nile tilapia.

O. niloticus is a freshwater fish species native to Africa found in rivers and streams in tropical region.  They are omnivorous, consuming phytoplankton, algae, detritus, and small invertebrates.

Channa punctatus is a freshwater fish species native to South and Southeast Asia found in rivers and streams in tropical region. They are carnivorous, feeding primarily on small fish, insects, crustaceans, and amphibians.

Barbonymus gonionotus is a freshwater fish native to Southeast Asia found in rivers and streams in tropical region. They are omnivorous, feeding on a variety of aquatic plants, algae, insects, and small invertebrates. They may also consume detritus and organic matter from the substrate.

Raiamas tornieri is a freshwater fish species found in rivers and streams in Southeast Asia. They are predator, feeding on smaller fish, aquatic insects, and crustaceans.

Anabas testudineus, a freshwater species, usually feed on insects, insect larvae, fry fishes, etc.

Monopterus albus is a freshwater  eel species found in various aquatic environments across East and Southeast Asia. They feed on small fish, insects, crustaceans, worms, and mollusks.

Clarias batrachus is a freshwater fish species native to Southeast Asia found in rivers and streams in tropical region. They are omnivorous, feeding on small fish, invertebrates, plant material, and detritus.

Channa striata is a freshwater fish species native to South and Southeast Asia found in rivers and streams in tropical region. They are carnivorous predator with a diet consisting primarily of smaller fish, crustaceans, amphibians, and insects.

Labeo rohita is a freshwater fish species found in rivers, lakes, and reservoirs in tropical region. They Feed on both plant matter and small aquatic organisms.

Hoplias intermedius, which are predatory freshwater fish found in South America. Preying on smaller fish, crustaceans, and other aquatic organisms. It likely inhabits slow-moving rivers, floodplains, and lakes, where it can take advantage of cover to hunt. 

Rewrite the Conclusion section. It is just a repetition of the main findings.
Edited in blue highlight  Page 15. Line 515-526. 

In the main text, the authors should choose how to refer a species, Nile tilapia or niloticus. Mixing names loses consistency. 
Edited in blue highlight Edited throughout the main text

Line 106:
Indicate which standard reference materials you used. American Public Health Association (2005). APHA Standard Methods Page 4. Line 162-163. 

Line 120: 
Indicate Bw. Is this average body weight of adults in Thailand?
Edited in blue highlight Page 5. Line 176-178. 

Line 132:
Delete "CR stands for cancer risk". 
Deleted. CR stands for cancer risk. Page 5. Line 188.

Lines 151-152: 
Delete this sentence.   (The statistical analysis of the genotoxicity study involved transferring each assessed DNA band to the dendrogram configuration utilizing the NTSYSpc 2.1 software [24].)  Page 6. Line 243.

Line 159:
Delete "(Table 3)" and write "in Table 3". 
Edited in blue highlight    In table 3. Page 7. Line 252.

Lines 206-207: Unclear. Rewrite this sentence.
Edited in blue highlight Page 8. Line 298-302.
Line 298: Delete this sentence.
 (The EDI, HRI and CR were used to assess the possible health risks associated with consuming HMs through fish. HRI represents health risk due to non-carcinogenic effects, while CR represents the health risk due to carcinogenic effects).

Reviewer 5 Report

Comments and Suggestions for Authors

The manuscript revision with recommendations for the authors is located in the Word file that has been uploaded.

Author Response

Introduction 
- Add a more detailed description of the research site, including information about the types of industries and the type of landfill located near the fish farm. 
Page 2. Line 46-50.
At the Khon Kaen municipal landfill adjacent to the waste-to-energy power plant, high concentrations of major heavy metals (HM) such as As, Cd, Cr, and Pb have been detested in both leachate and local organisms [3-7]. The reference site in the Thapra commercial fish farm situated at the green area of Khon Kaen city, without any industry and factory nearby. 

- It is necessary to include a more detailed description of the biological and ecological characteristics of T. niloticus.
Page 2. Line 68-71.
Oreochromis niloticus is a freshwater fish that primarily inhabits lakes, rivers, and reservoirs. As an herbivorous species, it grazes on periphyton, algae, and biofilms. The fish is highly susceptible to heavy metal pollution and other environmental contaminants, making it an important bioindicator of water quality. As a key protein source for human, O. niloticus is widely farmed in tropical and subtropical regions worldwide [10].

Materials and Methods
- The manuscript is missing data on the number of fish specimens collected for heavy metal analysis.
Edited. Page 3. Line 102-105. Water, sediment and O. niloticus were randomly selected from each of the 5 spots at the landfill site and the reference site for heavy metal analysis, genetic analysis and oxidative stress analysis (n=5 of each analysis, of each research sites) [4-5].    
-The number of specimens for genetic analysis and the number of specimens for oxidative stress analysis.    
Edited.    Page 3. Line 103-105.
- It is also necessary to specify the sediment sampling method.
Edited. Page 3. Line 115-118.
-And the number of samples taken in order to assess the representativeness of the sampling.    
Edited.    Page 3. Line 103-105.
-It is necessary to add the method of dissection of specimens when collecting tissue. 
Edited.    Page 3. Line 121-123.
- As well as the method of tissue preservation until chemical analysis.
Edited in grey highlight. Page 3. Line 119-125.
-The manuscript should specify the type of sample preparation for meat, water, and sediment. 
Edited in grey highlight. Line 109-125.
- Manual or using a microwave oven.     
Use manual. Line 131-163.
-The name of the ICP-OES device manufacturer and the purity of the reagents used.    
Edited in grey highlight. Page 4. Line 135-136.
-For the reliability of the results, it is advised that the authors provide the following information: which Reference material they used when achieving an accuracy range of 85-115% for determining heavy metal concentrations.
Edited in grey highlight. Page 4. Line 162-163.
The value of metals recovery is determined based on acceptance criteria that fall within the range of 85% to 115% (USEPA 1994) [18].    
-The wavelengths, and the detection limits for the metals they measured. Edited in grey highlight. Page 4. Line 158-160.
-The process of sample preparation for obtaining MDA, H2O2, SOD, and CAT is insufficiently described.    
Edited in grey highlight.    Page 6. Line 210-240.
Results
- Why was BAF calculated from tissue and water 
and why was the concentration of heavy metals in sediment not applied? Please provide an explanation.
Page 4-5. Line 164-171.
The Bioaccumulation Factor (BAF) for Oreochromis niloticus is typically calculated from tissue and water concentrations because BAF is a direct measure of how much of a substance (HMs) accumulates in an organism from its surrounding environment, particularly the water column. This helps assess the potential risk of heavy metal contamination in aquatic food webs and human consumption.

Reasons for Using Tissue and Water Concentrations:
Direct Uptake from Water: Fish can absorb heavy metals directly from the water through gills and skin, making water a crucial medium for bioaccumulation.

Standardized Methodology: Regulatory agencies and environmental assessments often rely on BAF calculations based on water concentrations because it allows for more consistent comparisons across different studies and species.

Relevance to Human Consumption: Measuring heavy metals in fish tissue
provides direct insight into potential health risks for humans who consume these fish.

Why Sediment Concentrations Were Not Applied:
Limited Direct Uptake: Although sediments act as a reservoir for heavy metals, fish do not typically absorb metals directly from sediments. Instead, they are exposed to metals that dissolve into the water or are bioavailable through their diet.    

Discussion 
-    This chapter needs to be supported by literature data regarding the transport and movement of heavy metals between water, sediment, and biota, with a particular focus on fish. 
Edited in grey highlight. Page 12. Line 354-358.
-    Additionally, it should specify which other toxic substances, besides heavy metals, can induce oxidative stress, along with a description of the mechanisms involved in this process.
Page 14. Line 456-459.
Edited. Several toxic substances besides heavy metals can induce oxidative stress by increasing the production of reactive oxygen species (ROS) or decreasing the body's ability to neutralize them. Some of these substances include:
Air Pollution, particulate Matter (PM), Ozone (O₃), and Nitrogen Dioxide (NO₂) can cause oxidative stress in the lungs and other organs when inhaled, leading to inflammation and cellular damage.
Cigarette Smoke including nicotine, tar, and carbon monoxide, which promote oxidative stress and contribute to chronic diseases like lung cancer, cardiovascular diseases, and respiratory disorders.
Alcohol, excessive alcohol consumption leads to the generation of ROS, mainly in the liver, contributing to liver damage, inflammation, and disorders like fatty liver disease and cirrhosis.
Pesticides and Herbicides, compounds like organophosphates and glyphosate can lead to oxidative damage in cells, contributing to neurological disorders, endocrine disruption, and even cancer.
Industrial Chemicals, such as benzene, toluene, and chlorinated solvents can induce oxidative stress in various tissues, increasing the risk of cancer and other degenerative diseases.
Oxidative stress occurs when there is an imbalance between reactive oxygen species (ROS) production and the body's antioxidant defenses. This imbalance can lead to cellular and tissue damage, contributing to various diseases such as cancer, neurodegenerative disorders, cardiovascular diseases, and aging.

-    After reviewing the literature data, explain the results of heavy metal concentrations for which statistically significant differences were found. Identify potential causes and consequences of these differences.    
Edited in grey highlight. Page 15. Line 498-511. Add reference [85-88]. 
Page 22-23.

Round 2

Reviewer 1 Report

Comments and Suggestions for Authors

The authors have adequately addressed my comments and made substantial revisions. I recommend the manuscript for publication in its current form.

Author Response

Thank you very much for your kind consideration.

Bundit

Reviewer 2 Report

Comments and Suggestions for Authors

.

Author Response

(The authors gave the same response as above.)

Reviewer 3 Report

Comments and Suggestions for Authors

Thank you for the additional work

It is a nice MS and shows clearly that aquaculture site has to be carefully evaluated 

this is an implication to highlight

Author Response

Response to Reviewer 3     (Pink highlight)

  Comments and Suggestions for Authors:

-Thank you for the additional work

-It is a nice MS and shows clearly that aquaculture site has to be carefully evaluated 

   this is an implication to highlight.

Response:  Authors edited as:

  • The reference site is an area where no contaminated leachate from industries, farms, or landfills (Lines 88–89).
  • This study showed that the parameter values of the reference site and the landfill site were within the range of the Thai surface water standards [34]. Monitoring water quality, including temperature, TDS, EC, DO and pH can support that these sites had potential impact on the suitability of the aquatic ecosystem [34,35]. Lines 344–347).
  • HMs (As, Cd, Cr, Pb) in water of the landfill reservoir exceeded Thailand's surface water standards. (Lines 517–518). 

 ….. However, CR value of Pb could pose a cancer risk factor. Heavy metals in the contaminated environment could induce genetic alterations in niloticus near the landfill. Fish from this site exhibited oxidative stress effects. (Lines 521–523).

Reviewer 4 Report

Comments and Suggestions for Authors

General comments

  • This study is of a local character.
  • My greatest concern is small sample size, 5 fish per sample site.
  • Information on sampling and sample preparation is unclear. For example, how is shoveling possible when the sampling sites are (if they are) deep? You had to use a grab sampler. How were the fish sacrificed and how you dried fish muscle? How can you measure oxidative stress biomarkers in the gills when you preserved this tissue in pure ethanol?

Specific comments

  • Lines 50-54: Rewrite this sentence. Avoid showing the numbers.
  • Lines 56-59: Remove to M&M.
  • Line 70: Unclear. An herbivorous species which feeds on insects?

Author Response

 Response to Reviewer 4  (Edited in Blue highlights)

   Comments and Suggestions for Authors:

   General comments

  • This study is of a local character.
  • My greatest concern is small sample size, 5 fish per sample site.

      Response:  

  • Water, sediment and niloticus were randomly selected from each of the 5 spots at the landfill and the reference sites for heavy metal analysis, genetic analysis and oxidative stress analysis (n=5; five samples of each analysis) [4,5]. [Lines 100–102]
  • For niloticus samples (5 fish per site with averagely 62- 85 g and age approximately 3 months), they were collected from each water site using the cast net fishing method. [Lines 115–116]

Description: The sample size was determined as five fish per sampling site, following the methodology adopted in previous studies, including Suttichaiya et al. (2016) [Ref. 4] and Phoonaploy et al. (2019) [Ref. 5]. 

  • Information on sampling and sample preparation is unclear. For example, how is shoveling possible when the sampling sites are (if they are) deep? You had to use a grab sampler.

           Response:  Authors edited at line 110-111.

For sediment samples, they were collected using a grab sampler from the bottom of the landfill reservoir, at a depth of approximately 40-50 cm below the water surface.

  • How were the fish sacrificed and how you dried fish muscle?

           Response: Authors edited at line 116-119.

The fish were humanly euthanized by overdose of tricane methane sulfonate (MS-222, Sigma-Aldrich, WA, USA). [line 116-118]

The tail muscle part was excised, dried in hot air oven at temperature 60ºC for 5 days and kept in room temperature [4,5] before measurement ….. [line 118-119]

  • Can you measure oxidative stress biomarkers in the gills when you preserved this tissue in pure ethanol?

            Response: ….. and the blood serum of each fish was kept in freezer at -80ºC    before measurements the levels of oxidative stress biomarkers …..   [line 121-122]

Specific comments

  • Lines 50-54: Rewrite this sentence. Avoid showing the numbers.

                    Response: Authors edited as  Swangneat et al. (2023) reported Cd, Cr and Pb contaminations in the muscle, gills, and liver of cage-cultured Oreochromis niloticus from a landfill area, with only Cd level exceeding the safety standard by 13% [3].  [line 52-54]

  • Lines 56-59: Remove to M&M.

                   Response: Authors removed to M&M  as  Consequences of acceptable limits HMs in life were reported by joint FAO/WHO expert committee on food additives (September 2002) were 0.01 mg/kg for As, 2mg/kg for Cd, 0.3 mg/kg for Hg in freshwater non-predatory fish, and 0.6 mg/kg for Hg in freshwater predatory fish.

         (move to  line 182-184) in Estimated daily intake (EDI) following the EDI formula

  • Line 70: Unclear. An herbivorous species which feeds on insects?

                   Response: Authors edited as  An herbivorous species, it feeds on algae and aquatic plants. [line 66]

Reviewer 5 Report

Comments and Suggestions for Authors

The authors have responded to the reviewers' requirements.

Author Response

(The authors gave the same response as above.)
